# Effects of Dietary Lactic Acid Supplementation on the Activity of Digestive and Antioxidant Enzymes, Gene Expressions, and Bacterial Communities in the Intestine of Common Carp, *Cyprinus carpio*

**DOI:** 10.3390/ani13121934

**Published:** 2023-06-09

**Authors:** Seyyed Morteza Hoseini, Morteza Yousefi, Alireza Afzali-Kordmahalleh, Esmaeil Pagheh, Ali Taheri Mirghaed

**Affiliations:** 1Inland Waters Aquatics Resources Research Center, Iranian Fisheries Sciences Research Institute, Agricultural Research, Education and Extension Organization, Gorgan 4915677555, Iran; esmaeilpaghe@gmail.com; 2Department of Veterinary Medicine, RUDN University, 6 Miklukho-Maklaya St., 117198 Moscow, Russia; myousefi81@gmail.com; 3Department of Aquatic Animal Health, Faculty of Veterinary Medicine, University of Tehran, Tehran 1417935840, Iran; afzali.alireza@ut.ac.ir (A.A.-K.); mirghaed@ut.ac.ir (A.T.M.)

**Keywords:** acidifier, intestinal microbes, intestinal immunity, hepatic antioxidant, aquaculture

## Abstract

**Simple Summary:**

Acidifiers such as lactic acid, formic acid, acetic acid, propionic acid, and citric acid are a class of feed supplements used in fish diets. Acidifiers play a role in the growth and development of intestinal villi and increase the absorption surface. Additionally, various studies have shown that these dietary supplements are utilized as nutrients by beneficial intestinal bacteria, which increase the populations of these bacteria and ultimately decreases the intestinal pH. The present study found that dietary 5 g/kg LA benefits common carp feeding to improve the growth rate, antioxidant capacity, and intestinal health. Such effects may be mediated via alterations in intestinal microbial communities.

**Abstract:**

The present study investigated the effects of dietary lactic acid (LA) supplementation on the growth performance, intestinal digestive/antioxidant enzymes’ activities, gene expression, and bacterial communities in common carp, *Cyprinus carpio*. Four diets were formulated to contain 0 g/kg LA (control), at 2.5 g/kg LA (2.5LAC), 5 g/kg LA (5LAC), and 10 g/kg LA (10LAC) and offered to the fish over a period of 56 days. The results showed that dietary 5 g/kg LA supplementation improved growth performance and feed efficiency in the fish. All LA treatments exhibited significant elevations in the intestinal trypsin and chymotrypsin activities, whereas the intestinal lipase, amylase, and alkaline phosphatase activities exhibited significant elevations in the 5LAC and 10LAC treatments. All LA treatments exhibited significant elevations in the intestinal heat shock protein 70, tumor necrosis factor-alpha, interleukin-1 beta, and defensin gene expressions, and the highest expression was observed in the 5LAC treatment. Additionally, dietary LA treatment significantly increased the lysozyme expression and *Lactobacillus* sp. population in the intestine of the fish, and the highest values were observed in the 5LAC and 10LAC treatments. *Aeromonas* sp. and *Vibrio* sp. populations decreased in the LA treatments, and the lowest *Aeromonas* sp. population was observed in the 10LAC treatment. The intestinal *mucin2* and *mucin5* expressions, and the hepatic reduced glutathione content, significantly increased, whereas hepatic glutathione peroxidase, glutathione reductase, and malondialdehyde significantly decreased in the 5LAC and 10LAC treatments. In conclusion, dietary 5 g/kg LA is recommended for common carp feeding to improve growth rate, antioxidant capacity, and intestinal health.

## 1. Introduction

With the expansion of the aquaculture industry, the need to adopt a suitable strategy to reduce rearing costs and increase feed efficiency has been increasingly noticed by aquaculture farmers [1]. Nowadays, the application of feed supplements is particularly popular among fish farmers due to the positive effects they have on fish growth and health [2,3]. Acidifiers such as lactic acid (LA), formic acid, acetic acid, propionic acid, and citric acid are examples of feed supplements used in fish diets [4,5,6]. Acidifiers play roles in the growth and development of intestinal villi and increase the absorption surface [7]. Additionally, various studies have shown that these dietary supplements are utilized as nutrients by beneficial intestinal bacteria, which increase the populations of these bacteria and ultimately decreases the intestinal pH [8]. In addition to facilitating the elimination of pathogens, the increase in intestinal acidity leads to the domination of beneficial microbes that can secrete digestive enzymes and degrade anti-nutrient factors, leading to better digestion and absorption [9,10]. Additionally, acidifiers are effective in increasing the secretion of mucins in the intestine [1,11], which, in addition to maintaining the health of digestive cells from scratches caused by feed, can increase the washing of pathogenic bacteria out of the digestive system of fish [12]. Organic acids are effective in reducing oxidative stress and increasing antioxidant capacity, which improves fish health and welfare [13,14,15]. In addition, by stimulating the immune system of fish, acidifiers increase the expression of some immune genes such as lysozyme (*lys*), interleukin-1 beta (*il1b*), tumor necrosis factor-alpha (*tnfa*), and defensin (*def*) play a role in fish resistance to diseases [5,16].

LA is one of the well-known acidifiers in the aquaculture industry, which is used as a feed supplement. Studies have shown that diets containing 10–20 g/kg of LA can increase the growth performance and digestibility of nutrients in beluga, *Huso huso* [17], and improve immunological and antioxidant markers in rainbow trout, *Oncorhynchus mykiss* [14]. Additionally, incorporating sodium lactate into the diet has increased the growth performance of narrow-clawed crab, *Astacus leptodactylus* [16], and Arctic charr, *Salvelinus alpinus* [18]. However, LA and its salt have shown no effects on weight gain and feed efficiency in rainbow trout [14] and Atlantic salmon, *Salmo salar* [19], suggesting that further studies are needed to illustrate the effects of LA in fish.

Common carp, *Cyprinus carpio*, is a highly sought-after fish in many countries (fourth most produced species) due to its firm and tasty flesh [20]. It is a hardy fish that can tolerate a wide range of environmental conditions, including low oxygen levels and high temperatures. Common carp has a fast growth rate and can be harvested within a relatively short time [21]. Considering the beneficial effects of acidifiers such as LA on fish, the expansion of their use in the aquaculture industry, and the lack of comprehensive and sufficient information about the effects of LA on common carp, this research aimed to investigate the effect of dietary LA supplementation on growth performance, hepatic antioxidant parameters, intestinal digestive enzymes, expression of immune-related genes, and bacterial populations in common carp.

## 2. Materials and Methods

### 2.1. Diets

Fish diets were made using available local feedstuffs and based on the nutritional requirement of common carp. The feedstuffs were weighed and mixed, and then 400 mL/kg water was added to the mixture to create a dough. LA was added at the expense of the water at 2.5 g/kg (2.5LAC), 5 g/kg (5LAC), and 10 g/kg (10LAC). A diet without LA supplementation was considered the control diet (CTL, Table 1). A meat grinder was used to create the feed pellets, which were dried against a fan blower. The diets were analyzed for crude protein (the Kjeldahl method), crude fat (ether extraction), crude ash (burning in a muffle furnace), and crude fiber (acid/base digestion) according to the standard methods [22].

### 2.2. Experimental Protocol

Two hundred and twenty common carp juveniles (~25 g) were purchased from a local farm and transferred to the laboratory. They were stocked in a 1500 L tank for ten days for acclimation, during which they were fed the CTL diet. After that, 180 healthy fish (with no external lesions and abnormalities) were randomly stocked in 12 tanks (150-L), 15 fish per tank. The tanks were equipped with aeration and a water flow rate of 0.3 L/min and divided into four triplicate groups, each considered as a treatment that fed either CTL, 2.5LAC, 5LAC, or 10LAC diet (3% of biomass divided in 3 meals per day) over 8 weeks. The tanks’ biomasses were recorded on day 0 (when the fish were stocked in the 150 L tanks) and every other week until the eighth week, and the feed amounts were corrected based on the biomasses. Water temperature (24.6 ± 0.59 °C), dissolved oxygen (6.87 ± 0.81 mg/L), pH (7.39 ± 0.41), salinity (2.52 ± 0.13 g/L), and total ammonia nitrogen (2.65 ± 1.02 mg/L) levels were measured during the rearing period. The fish were reared under a natural photoperiod (14 L/10 D). At the end of the experiment, specific growth rate (SGR), weight gain, and feed efficiency were determined based on the following formulas [23]:$$\mathrm{SGR}\;(\%/d)=100\;\times \;\frac{Ln(final\;weight)-Ln(initial\;weight)}{56}$$
$$\mathrm{Weight}\;\mathrm{gain}\;(\%)=100\;\times \;\frac{Final\;weight(g)-Initial\;weight(g)}{Initial\;weight(g)}$$
$$\mathrm{Feed}\;\mathrm{efficiency}\;(\%)=\frac{Gained\;biomass(g)}{Consumed\;feed(g)}$$

### 2.3. Sampling and Preservation

At the end of the experiment, the intestine and liver samples were collected from all treatments. After 24 h of fasting, three fish were caught from each tank and immediately anesthetized in a eugenol bath (100 mg/L) and then euthanized by a sharp blow on the head. The abdominal cavity of fish was opened by scissors and a piece of the liver was dissected and frozen in liquid nitrogen for antioxidant assays. The anterior intestine of the fish was dissected and frozen in liquid nitrogen for digestive enzymes’ assays. The posterior intestine was dissected and frozen with the chyme in liquid nitrogen for gene expression and bacterial population examination.

#### 2.3.1. Hepatic Antioxidant Assays

The hepatic samples were homogenized in a mortar with liquid nitrogen. Then, phosphate buffer was added to the homogenate (1:1 ratio) and mixed thoroughly. The mixture was then centrifuged at 4 °C for 30 min (13,000× *g*). The supernatant was collected in a new tube and used for soluble protein assays, according to Bradford [21]. The remaining extract was used for reduced glutathione (GSH), glutathione peroxidase (GPx), glutathione reductase (GR), and malondialdehyde (MDA) assays using commercial kits (Zellbio GmbH Co., Deutschland, Germany). GSH reaction with 5,5′-dithiobis-(2-nitrobenzoic acid) was used to assay the extract GSH content at 412 nm. GPx and GR activities were measured based on the conversion of GSH to oxidized glutathione (GSSG) by GPx and the recycling of GSSG by GR that consumes NADPH. The decrease in the NADPH content was measured at 340 nm. The MDA concentration was determined after deproteinization with tricarboxylic acid and reaction with thiobarbituric acid in the presence of butylated hydroxytoluene at 95 °C.

#### 2.3.2. Digestive Enzymes’ Assay

The intestine samples were homogenized with three volumes of phosphate buffer (pH 7.0) in a mortar and in the presence of liquid nitrogen. After 2 min of homogenizing, the homogenate was poured into 3 separate 2 mL tubes and centrifuged for 30 min at 4 °C (13,000× *g*). The supernatants were collected and preserved at −70 °C for further analysis. Total soluble protein was measured using the Bradford method [24]. Trypsin activity was measured at 410 nm using DL-arginine-p-nitroanilide as the substrate, according to a previous report [25]. N-benzoyl-L-tyrosine ethyl ester was used as the substrate for the determination of chymotrypsin activity at 256 nm, according to a previous report [26]. Amylase activity was measured according to the method described previously [27], using soluble starch as the substrate. The increase in the reducing power of buffered starch solution was measured with 3–5 dinitro salicylic acid (DNS) at 540 nm. Lipase activity was determined based on the hydrolysis of p-nitrophenyl myristate at 37 °C and wavelength 580 nm [28]. Alkaline phosphatase (ALP) activity was determined based on the hydrolysis of the p-nitrophenyl phosphate to p-nitrophenol and phosphate in the presence of magnesium ions, as described previously [29].

#### 2.3.3. Intestinal Gene Expression

RNA was extracted from the intestine samples using a commercial kit (Denazist Co., Tehran, Iran) and treated with DNase I (Thermo Fisher Scientific, Waltham, MA, USA) to remove any DNA contaminations. Then, a commercial kit (SMOBIO Technology Co.; Hsinchu City, Taiwan) was used to synthesize cDNA, and gene expression was determined based on a quantitative RT-PCR using an instrument supplied by Applied Biosystem (Waltham, MA, USA). Specific primers of *lys*, heat shock protein 70 (*hsp*), *tnfa*, *il1b*, *def*, mucin2 (*muc2*), and mucin5 (*muc5*) for common carp (Table 2) and a SYBR Green kit (Ampliqon A/S, Stenhuggervej 22, Odense M, Denmark) were used for qRT-PCR. *beta-actin* was used as the housekeeping gene, and the expression of the target genes was determined based on Livak and Schmittgen [30].

#### 2.3.4. Intestinal Bacterial Genus Populations

Populations of *Lactobacillus* sp., *Aeromonas* sp., and *Vibrio* sp. were examined in the intestinal samples. The samples were digested, and their DNA was extracted via the phenol-chloroform method using a washing kit provided by GeneAll Co. (Seoul, Korea) as described previously [14]. Specific primers were designed for the target bacteria groups and the universal 16 s primer was used for an examination of the total bacteria population (Table 3). qRT-PCR was used for amplification, and bacterial populations were calculated based on the ∆∆Ct method [14].

### 2.4. Statistical Analysis

After confirming normal distribution (Shapiro–Wilk’s test) and homoscedasticity (Levene’s test), the data were analyzed using a one-way ANOVA and Duncan tests at a significance level of 0.05. The analyses were conducted in SPSS v.22 (IBM corporation, Chicago, IL, USA), and the data were presented as mean ± SE.

## 3. Results

There was no mortality during the experiment. Final weight, SGR, weight gain, and feed efficiency in the 5LAC treatment were significantly higher than those of the CTL and 10LAC treatments. There were no significant differences in these parameters between the 2.5LAC and 5LAC treatments (Table 4).

All LA treatments showed significant rises in the intestinal trypsin and chymotrypsin activities compared to the CTL group; however, no significant differences were observed among the 2.5LAC, 5LAC, and 10LAC treatments (Figure 1). The intestinal lipase, amylase, and ALP activities exhibited significant elevations in the 5LAC and 10LAC treatments compared to the CTL. The activities of these enzymes in the 2.5LAC treatment were similar to those of the CTL, 5LAC, and 10LAC treatments (Figure 1).

As all the LA treatments exhibited significant diminutions in the hepatic MDA contents, the lowest value was related to the 5LAC and 10LAC treatments (Figure 2). There were no significant differences in the hepatic GPx and GR activities and the GSH content between the 2.5LAC and CTL treatments; nevertheless, these treatments had significantly lower hepatic GPx and GR activities and higher GSH content compared to the CTL treatment (Figure 2).

The intestinal expressions of *hsp*, *tnfa*, *il1b*, *def*, and *lys* genes significantly up-regulated in the LA treatments compared to the CTL treatment (Figure 3). The highest expressions of the intestinal *hsp*, *tnfa*, *il1b*, and *def* were observed in the 5LAC treatment; the highest expression of the intestinal *lys* was observed in the 5LAC and 10LAC treatments. While there were no significant differences in the intestinal *muc2* expression between the CTL and 2.5LAC treatments, the 5LAC and 10LAC treatments exhibited significant up-regulations in the expression of this gene compared to the CTL treatment. An increase in dietary LA levels significantly increased the intestinal *muc5* expression. The highest intestinal expression of *muc2* was observed in the 5LAC treatment; the highest expression of the intestinal *muc5* was observed in the 10LAC treatment (Figure 3).

Dietary lactic acid supplementation significantly decreased the populations of *Aeromonas* sp. and *Vibrio* sp. and increased the population of *Lactobacillus* sp. in the fish intestine (Figure 4). The lowest intestinal *Aeromonas* sp. population was observed in the 10LAC treatment, but there was no significant difference between the 2.5LAC and 5LAC treatments. The intestinal *Vibrio* sp. were similar among the 2.5LAC, 5LAC, and 10LAC treatments. The highest intestinal *Lactobacillus* sp. population was observed in the 10LAC treatment, which was statistically similar to the 5LAC treatment.

## 4. Discussion

Organic acids have become increasingly popular as feed additives in aquaculture due to their benefits, such as improved nutrient digestibility, growth performance, immune stimulation, and intestinal health [35]. The effectiveness of organic-acid-supplemented diets in promoting growth performance in fish is not always consistent. Studies have shown that the use of LA or its salt has resulted in varying outcomes. For example, the addition of 20 g/kg of LA to fishmeal- and plant-based diets has led to improved growth performance in beluga [17], while 16.7 g/kg of sodium lactate supplementation has not promoted growth performance in giant grouper, *Epinephelus lanceolatus*, fed a plant-based diet [36]. Rainbow trout has exhibited no growth promotion when fed diets containing 5–20 g/kg LA [14]. In crustaceans, supplementation with 5–50 g/kg of encapsulated sodium lactate has significantly improved the growth performance of narrow-clawed crayfish [16]. These inconsistencies may be due to species-specific traits. Supporting this, research comparing Arctic charr and Atlantic salmon under similar experimental conditions found that dietary sodium lactate supplementation (10 g/kg) was beneficial for Arctic charr but not Atlantic salmon. The authors attributed this difference to the longer gastric emptying time in Arctic charr, which reduces harmful bacterial populations in the intestine [18].

One of the benefits of organic acid supplements is their ability to improve nutrient digestibility, which can increase growth performance in fish [16,37,38]. Similarly, LA supplementation has been shown to increase pancreatic enzymes’ activities in the fish intestine, as seen in previous studies on other fish species treated with other organic acids [37,38]. Studies have suggested that organic acids increase the activity of pancreatic enzymes in the fish intestine [37,38,39,40]. However, it is not clear if such increases result from the higher secretion of the pancreatic enzymes to the lumen and/or they are derived from the microbes of the lumen. Barlaya et al. [41] have shown that increase in a plant protein source in fish diet decreases the activity of digestive enzymes in the intestine, but not hepatopancreas. Moreover, organic acids may have no significant effects on digestive enzymes in the pancreas, as evidenced in shrimp [42,43]. It has been suggested that pancreatic enzymes are sensitive to anti-nutrients present in plant proteins, and significant decreases in intestinal digestive enzymes have been reported under such situations [44,45]. For example, phytate, a well-known anti-nutrient in plants, can inhibit lipase, amylase, and protease activity [46]. Beneficial bacteria can secrete digestive enzymes in the fish intestine [9]. Moreover, they can degrade anti-nutrients and suppress their negative effects on digestive enzymes. For example, many lactic acid bacteria produce phytase that can breakdown dietary phytate [10], probably improving digestive enzymes’ activity. Therefore, it is speculated that dietary LA increased the intestinal activity of amylase, lipase, trypsin, and chymotrypsin by altering the intestinal microbiota. Interestingly, the present study also found that LA supplementation increased intestinal ALP activity. ALP is a brush-border enzyme responsible for nutrient absorption [45]. ALP elevation may be due to an increase in enzyme production and/or intestinal folding, resulting in a higher surface-to-weight ratio of the intestine [47]. Several articles have reported results which support the claim that organic acids increase intestinal folding in fish [48,49,50].

Oxidative stress is one of the most life-threatening events in fish because of the high concentration of unsaturated fatty acids in the fish body [51]. Elevation in pro-oxidant agents in the fish bodies results in the activation of the antioxidant system, characterized by elevation in antioxidant enzymes’ activity to neutralize the harmful compounds [52]. If the antioxidant capacity fails to fully clear the harmful agents, they attack biological molecules, particularly fatty acids. MDA is a product of lipid peroxidation, and a low MDA concentration is an indicator of lower oxidative condition [53]. GSH-dependent antioxidant factors play a significant role in protecting fish cells against oxidation [54]. GPx uses GSH as a co-factor to neutralize hydrogen peroxide and other hydroperoxides. As a result, GSH is oxidized and loses its biological functions. GR is responsible for reducing oxidized glutathione. Dietary LA supplementation increased GSH levels and decreased GPx and GR activities, accompanied by lower MDA levels [55]. Hence, this study suggests that LA improved the antioxidant capacity of the intestine, leading to less GSH oxidation, thus lower activity of the glutathione-related enzymes. It has been reported that dietary organic acid/salt can increase antioxidant capacity in fish. Although there are limited data on this topic, dietary LA supplementation has been shown to significantly increase antioxidant enzymes’ activity and decrease MDA concentration in fish [14,56]. The antioxidant-modulating effect of dietary LA supplementation may be associated with an improvement in the fish intestinal microbiota, as studies have shown that beneficial bacteria can enhance antioxidant capacity in fish [38,57,58]. Specifically, it has been reported that some *Lactobacillus* sp. are capable of producing GSH [59,60] and riboflavin [61], which is needed for the GSH redox cycle [62,63]. Therefore, it is speculated that the increase in the population of *Lactobacillus* sp. bacteria in the fish intestine has improved the antioxidant capacity and reduced lipid peroxidation in the present study.

Organic acids can change the intestinal microbiota by reducing intestinal pH and serving as nutrients for certain bacteria. After the utilization of a specific organic acid/salt, other metabolites are released by the host intestinal microbes that serve as nutrients for other species, potentially changing the entire microbiota of the fish intestine [64]. Numerous studies have demonstrated that dietary butyrate [49,65,66], acetate [67], or LA [14] increases the population of *Lactobacillus* sp. in the fish intestine, which is consistent with the present results. In addition, the present study showed that dietary LA supplementation subordinates the populations of harmful bacteria, which is consistent with previous studies that showed that organic acids decreased *Streptococcus iniae* [14], *Pseudomonas* sp. [68], and *Vibrio* sp. [69] in the fish intestine. Overall, the present study suggests that dietary LA supplementation may be an effective procedure to balance the intestinal microbial populations in common carp, which can be the main reason for the benefits of LA in this species.

The intestine plays a crucial role in the immune function of fish as it serves as a barrier against harmful pathogens. The interaction between the intestinal microbiota and antigens with the host fish occurs at the gastrointestinal mucosal surface [70]. Fish rely on defensins for various immune-related functions, such as antibacterial, antiviral, and anti-inflammatory roles [71]. The intestinal *def* expression has been up-regulated by dietary feed additives and is known as improved immunity of the fish intestine [72,73,74]. Data regarding the roles of organic acids on fish intestinal *def* expression are scarce; dietary LA supplementation has had no significant effects on *def* expression in the intestine of rainbow trout [14]. The present study suggests that dietary LA administration is an effective practical approach for increasing the intestinal *def* expression in common carp, which may be helpful in strengthening intestinal immunity and health.

Heat shock protein 70 is commonly recognized for safeguarding living cells from stressful conditions by maintaining proper protein folding and function [75]. The expression of *hsp* can vary under different conditions; for instance, handling stress did not affect *hsp* expression in rainbow trout [76] but decreased the gene expression in Yellow Perch, *Perca flavescens* [77]. Similarly, dietary LA [14] and malic acid [78] supplementation has been reported to up-regulate the intestinal *hsp* expression in the intestine of rainbow trout. These up-regulations in intestinal *hsp* expressions have been in line with the domination of beneficial bacteria or boosted the antioxidant capacity of the fish intestine. Therefore, it is assumed that the increased intestinal *hsp* expression in LA-treated fish was a positive response that enhanced the cell’s ability to handle potential stressors, which suggests that the up-regulation of the *hsp* expression in the present study had been a sign of boosted intestinal health.

Fish possess a diverse group of glycoproteins known as mucins, which are present in the mucosal barrier. Among these, mucin2 and mucin5 are significant macromolecules that create a gel layer. Studies on humans [79] and broilers [80] have revealed that short-chain fatty acids may induce signals to increase mucins. Data regarding the effects of organic acids on fish intestinal *mucin* expression are scarce, as an increase in *mucin* expression in the intestine of gilthead sea bream has been reported after butyrate administration [81]. Fish intestinal mucins are crucial for immune responses and have been observed to change during infections [82,83] or exposure to toxicants [12,84]. It has been found that fish intestinal mucins are depleted within the first days after bacterial/parasitic infections [84,85], a defense mechanism to washing out pathogens, but recover after that. On the other hand, viral infection in common carp has been found to down-regulate *mucin* expression, which is a sign of distress and susceptibility to subsequent infections [86]. So, an increase in intestinal *mucins* can be considered a protective response that may help the fish for subsequent pathogenic challenges.

Immune cells produce cytokines such as il1b and tnfa which initiate the defense mechanism of the immune system by enhancing phagocytosis, respiratory burst activity, and nitric oxide production [87]. Lysozymes are essential molecules for innate immunity that exhibit a strong catalytic ability to break down bacteria cell walls, promoting cell lysis in the hypoosmotic environment [88]. With its antibacterial, antiviral, and anti-inflammatory properties, lysozyme can activate nuclear leucocytes and macrophages to promote phagocytosis [89]. The expression level of *lys*, *il1b*, and *tnfa* are important indicators for monitoring fish immunity and evaluating the effects of diet and vaccines. Similar to the present results, dietary malic acid [78] and butyric acid [66] have increased the expression of *lys*, *il1b*, and *tnfa* in the fish intestine. It has been found that induction of *lys*, *il1b*, and *tnfa* expression in the intestine of fish after the administration of organic acids has been accompanied by high disease resistance [66,87,90]. Interestingly, such an improvement in disease resistance has been accompanied by the domination of beneficial bacteria and subordination of harmful bacteria in the intestine of crucian carp, *Carassius auratus gibelio* [90], which can partly explain the present results.

## 5. Conclusions

In conclusion, the present study demonstrates that dietary supplementation with lactic acid improves growth performance and feed efficiency in fish by enhancing intestinal enzyme activities, regulating gene expressions related to immune response, increasing beneficial bacterial populations, and providing antioxidant benefits. It seems that such benefits are mediated by the domination of beneficial bacterial communities in the intestine. Although most of the tested parameters suggest that 5 g/kg LA is suitable for common carp dietary supplementation, further studies are needed to address the reasons behind the difference in the responses of some parameters to dietary LA levels.

## Figures and Tables

**Figure 1 animals-13-01934-f001:**
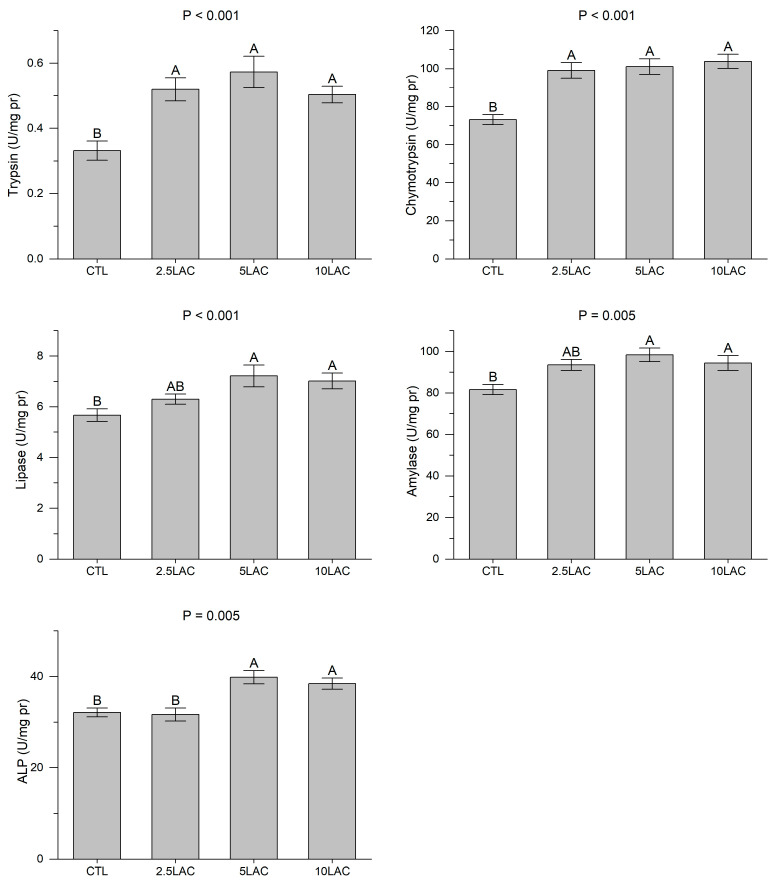
The intestinal activities of the digestive enzymes in common carp fed diets supplemented with 0 g/kg (CTL), 2.5 g/kg (2.5LAC), 5 g/kg (5LAC), and 10 g/kg (10LAC) lactic acid. Significant differences among the treatments were indicated by different capital letters above the bars (*n* = 3; Duncan).

**Figure 2 animals-13-01934-f002:**
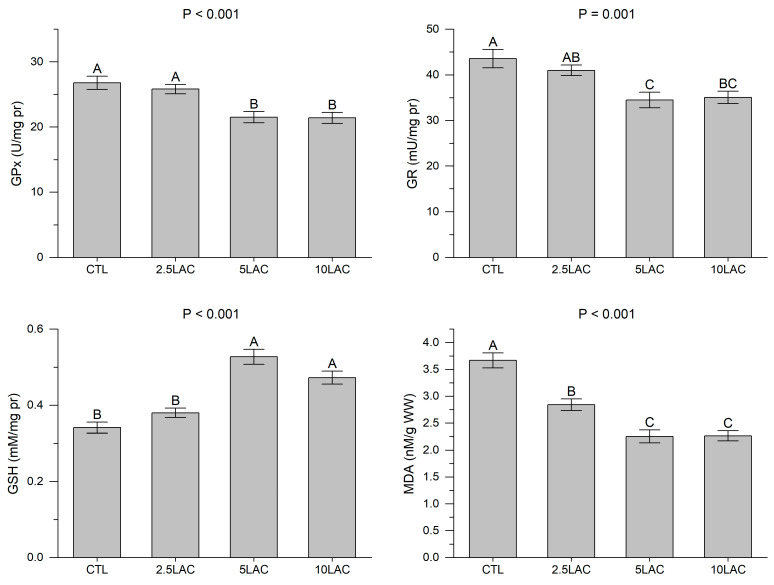
The hepatic antioxidant parameters in common carp fed diets supplemented with 0 g/kg (CTL), 2.5 g/kg (2.5LAC), 5 g/kg (5LAC), and 10 g/kg (10LAC) lactic acid. Significant differences among the treatments were indicated by different capital letters above the bars (*n* = 3; Duncan).

**Figure 3 animals-13-01934-f003:**
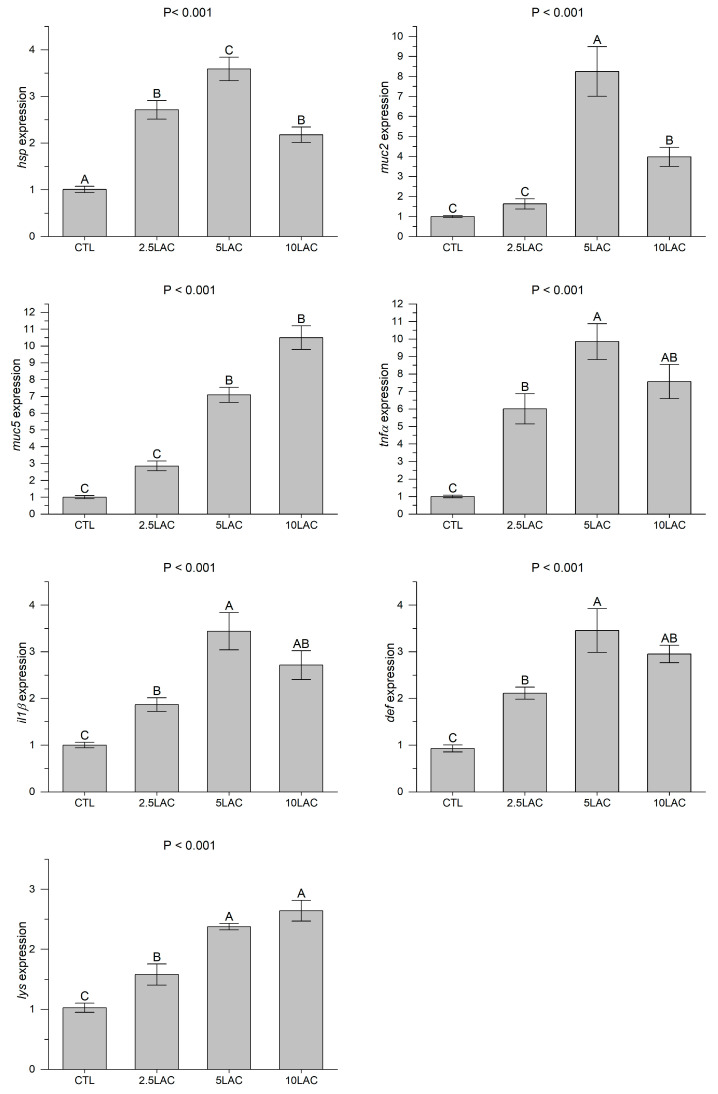
The intestinal gene expressions (relative to beta-actin) in common carp fed diets supplemented with 0 g/kg (CTL), 2.5 g/kg (2.5LAC), 5 g/kg (5LAC), and 10 g/kg (10LAC) lactic acid. Significant differences among the treatments were indicated by different capital letters above the bars (*n* = 3; Duncan).

**Figure 4 animals-13-01934-f004:**
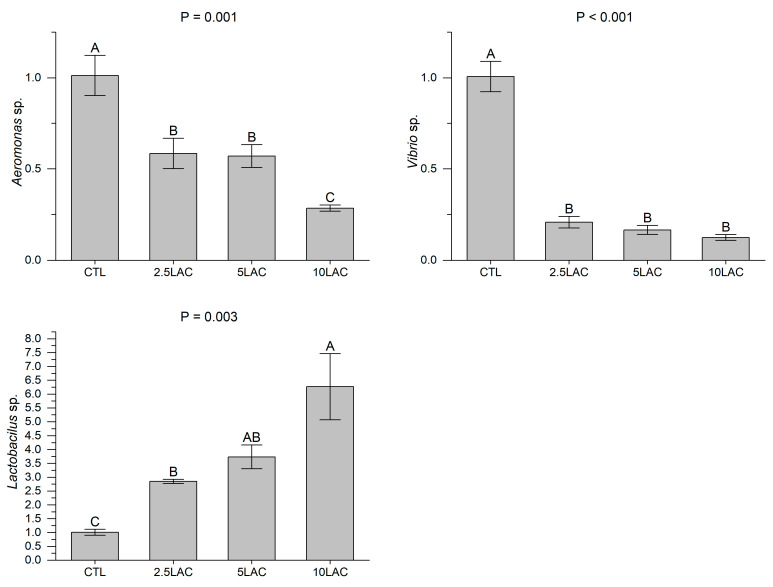
The intestinal populations of *Aeromonas* sp., *Vibrio* sp., and *Lactobacillus* sp. in common carp fed diets supplemented with 0 g/kg (CTL), 2.5 g/kg (2.5LAC), 5 g/kg (5LAC), and 10 g/kg (10LAC) lactic acid. Significant differences among the treatments were indicated by different capital letters above the bars (*n* = 3; Duncan).

**Table 1 animals-13-01934-t001:** Dietary ingredients and proximate composition.

	Dietary Lactic Acid Levels (g/kg)			
	0 (CTL)	2.5 (2.5LAC)	5 (5LAC)	10 (10LAC)
Corn meal ^1^	100.0	100.0	100.0	100.0
Wheat flour ^2^	280.0	277.5	275.0	270.0
Soybean oilcake ^3^	200.0	200.0	200.0	200.0
Poultry by-product ^4^	380.0	380.0	380.0	380.0
Plant oil (corn oil + sunflower oil; 1:1 ratio)	20.0	20.0	20.0	20.0
Vitamin + Mineral premix ^5^	15.0	15.0	15.0	15.0
Methionine ^6^	3.0	3.0	3.0	3.0
Lysine ^7^	2.0	2.0	2.0	2.0
Lactic acid ^8^	0.0	2.50	5.0	10.0
Proximate composition (g/kg)				
Moisture	88.6	87.4	87.3	87.9
Crude protein	381	384	386	379
Crude fat	105	108	105	107
Crude ash	53.4	54.0	53.4	55.0
Crude fiber	42.1	42.0	41.0	42.3
Crude energy (kcal/g)	3789	3806	3798	3794

^1^. Containing 8.9%, 3.5%, 5.3%, and 2.6% of crude protein, fat, ash, and fiber, respectively. ^2^. Containing 11.1%, 1.5%, 2.3%, and 2.5% of crude protein, fat, ash, and fiber, respectively. ^3^. Containing 44.3%, 1.88%, 5.32%, and 3.68% of crude protein, fat, ash, and fiber, respectively. ^4^. Containing 63%, 16%, 6%, and 4% of crude protein, fat, ash, and fiber, respectively. ^5^. Supplied by Amineh Gostar Co. (Tehran, Iran), providing per kg if diet: B2: 10 mg; E: 20 mg; K: 24 mg; B3: 12 mg; B5: 40 mg; B6: 5 mg; B1: 4 mg; A: 1600 IU; D3: 500 IU; H: 0.2 mg; B9: 2 mg; B12: 0.01 mg; C: 60 mg; Inositol: 50 mg; Iodate: 0.05 mg; Fe: 2.5 mg; Co: 0.04 mg; Cu: 0.5 mg; Zn: 6 mg; Choline: 150 mg; Se: 0.15 mg; Mn: 5 mg. ^6^. CJ CheilJedang Corporation, Seoul, South Korea. ^7^. CJ CheilJedang Corporation, Seoul, South Korea. ^8^. Purity: 85%; food grade; supplied by Mobtakeran Shimi Corporation, Tehran, Iran.

**Table 2 animals-13-01934-t002:** The sequence of the target genes’ primers (with the primer length, melting temperature, amplicon length, and accession number).

Primer		Sequence (5-3)	Length	Tm	Amplicon (bp)	Accession No.	Amplification Efficiency
*hsp*	F	ATGTTGCCTTCACAGACACTG	21	60	120	XM_042720446.1	1.67
	R	GGTCATCAAACTTTCTGCCGA	21	60			
*muc2*	F	ATTGGCATTGAGTTCACCGAG	21	60	135	XM_042752573.1	1.86
	R	GACAGTGATGCCCATTTTGGA	21	60			
*muc5*	F	TGTGTGAGCATGGGGTGTATA	21	60	141	XM_052598245.1	1.85
	R	CTGTTGAACTTGCTCTCCAGG	21	60			
*lys*	F	CAGGTGGAAAGAACAAGTGCA	21	60	150	XM_019104788.1	1.78
	R	ACATCTTACGCCCCTTACAGT	21	60			
*def*	F	GCAAAGAGAATGAGGCTGTGT	21	60	132	JF343439.1	1.84
	R	CACAGCACAAAAATCCCTTGC	21	60			
*tnfa*	F	GAACAATCAGGAAGGCGGAAA	21	60	128	XM_019088899.1	1.69
	R	GGGTTTCTGTGGACACTTCAG	20	60			
*il1b*	F	CATTGCTTGTACCCAGTCTGG	21	60	121	XM_042733144.1	1.82
	R	TCTGAAGAAGAGGAGGCTGTC	21	60			
*beta-actin*	F	TCTGCTATGTGGCTCTTGACT	21	60	118	XM_042721308.1	1.94
	R	AACCTCTCATTGCCAATGGTG	21	60			

**Table 3 animals-13-01934-t003:** The primer sequences used for detecting different bacteria in the fish intestine.

Bacterium	Name	Sequences	Reference
*Lactobacillus* sp.	Lacto-F	TGGAAACAGRTGCTAATACCG	[31]
	Lacto-R	GTCCATTGTGGAAGATTCCC	
*Vibrio* sp.	Vibrio-F	GGCGTAAAGCGCATGCAGGT	[32]
	Vibrio-R	GAAATTCTACCCCCCTCTACAG	
*Aeromonas* sp.	Aeromonas-F	GAGAAGGTGACCACCAAGAACA	[33]
	Aeromonas-R	CTGACATCGGCCTTGAACTC	
All bacteria	338F	ACTCCTACGGGAGGCAGCAG	[34]
	518R	ATTACCGCGGCTGCTGG	

**Table 4 animals-13-01934-t004:** Growth performance, feed efficiency, and survival of common carp fed diets supplemented with 0 g/kg (CTL), 2.5 g/kg (2.5LAC), 5 g/kg (5LAC), and 10 g/kg (10LAC) lactic acid. Significant differences among the treatments were indicated by different superscript capital letters within a row (*n* = 3; Duncan).

	CTL	2.5LAC	5LAC	10LAC	P
Initial weight (g)	25.8 ± 0.09	25.8 ± 0.11	25.7 ± 0.04	25.8 ± 0.09	0.987
Final weight (G)	45.8 ± 0.65 ^B^	47.2 ± 0.69 ^AB^	49.7 ± 0.96 ^A^	46.0 ± 0.83 ^B^	0.029
SGR (%/d)	1.03 ± 0.02 ^B^	1.07 ± 0.03 ^AB^	1.17 ± 0.04 ^A^	1.03 ± 0.03 ^B^	0.031
Weight gain (%)	77.7 ± 2.26 ^B^	83.1 ± 3.50 ^AB^	93.6 ± 4.08 ^A^	78.2 ± 3.07 ^B^	0.031
Feed efficiency (%)	0.58 ± 0.02 ^B^	0.60 ± 0.01 ^AB^	0.65 ± 0.02 ^A^	0.58 ± 0.01 ^B^	0.028
Survival (%)	100	100	100	100	1.00

## Data Availability

The data of this study are available upon request from the corresponding author.

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
