# Peer review of "Effects of Dietary Lactic Acid Supplementation on the Activity of Digestive and Antioxidant Enzymes, Gene Expressions, and Bacterial Communities in the Intestine of Common Carp, Cyprinus carpio"

_animals, 2023, doi:10.3390/ani13121934_

Round 1

Reviewer 1 Report

Dear authors,

As requested, I reviewed the manuscript “Effects of dietary lactic acid supplementation on intestinal digestive enzymes, antioxidant activity, gene expression and bacterial community in common carp, Cyprinus carpio” by Hoseni SM, Yousefi M, Kordmahaleh AA., Pagheh E and Mirghaed AT. The work focused on the administration of different concentrations (0, 2.5, 5 and 10 g/kg) of an organic acid (lactic acid) on common carp for 56 days. The analysis included growth performance, digestive enzymes, antioxidant activity, gene expressions and bacterial community. The results are nice and look interesting, supported by an important sampling scheme. However, the work shows some weaknesses that need to be improved and, for this, I suggest to accept the paper with minor revisions, which are as follows:

1.     The Introduction looks clear and nice, but the authors never state why common carp has been chosen for the analysis. The advantages of this fish in the field of aquaculture should be mentioned;

2.     In Material and Methods there are few imperfections and some missing information. For example, the authors should state which local feed has been administered as control diet (commercial or laboratory diet). More detailed information (like temperature, light/dark cycle, salinity, etc) about animal rearing should be added. Furthermore, the authors should clarify the number of individuals per tank and that the results are expressed as the mean of the three replicates, each derived from a pool of 15 samples (if I understand correctly). Likewise, the formulas used for growth performance should be expressed as the mean of three replicates, each derived from a pool of 15 samples. Moreover, reference(s) should be added for the formulas. The sentence “two fish were caught for each aquaria and immediately anesthetized…” at line 124 should be changed: it is as if only two fish per group were sampled. Finally, references used for the protocols used in “2.4. Digestive enzymes’ assays” are too old and should be updated;

3.     Most of the accession numbers shown in Table 1 has been removed on the gene bank of NCBI;

4.     The results of gene expression do not seem to be expressed as 2-∆∆CT as stated in Materials and Methods;

5.     Finally, the Discussion is the real weak point of the paper and it should be rewritten properly. The authors never discuss the growth performance results and most of digestive enzymes and antioxidant activities results. But above all, the authors never discuss the results on the basis of the different concentrations administered. It is clear that some data show an increase concentration dependent while others show a decrease from a concentration onward and this should be must be taken into account without generalizing.

Thank you very much for your attention to my opinion.

The English language needs to be revised.

Author Response

Reviewer 1

As requested, I reviewed the manuscript “Effects of dietary lactic acid supplementation on intestinal digestive enzymes, antioxidant activity, gene expression and bacterial community in common carp, Cyprinus carpio” by Hoseni SM, Yousefi M, Kordmahaleh AA., Pagheh E and Mirghaed AT. The work focused on the administration of different concentrations (0, 2.5, 5 and 10 g/kg) of an organic acid (lactic acid) on common carp for 56 days. The analysis included growth performance, digestive enzymes, antioxidant activity, gene expressions and bacterial community. The results are nice and look interesting, supported by an important sampling scheme. However, the work shows some weaknesses that need to be improved and, for this, I suggest to accept the paper with minor revisions, which are as follows:

  1. The Introduction looks clear and nice, but the authors never state why common carp has been chosen for the analysis. The advantages of this fish in the field of aquaculture should be mentioned;

Response: Thank you. A paragraphs describing the benefits of common carp was added to the introduction.

  1. In Material and Methods there are few imperfections and some missing information. For example, the authors should state which local feed has been administered as control diet (commercial or laboratory diet).

Response: Please be noted that we used local feedstuffs, not local diet. The list of feedstuffs is presented in the Table 1.

  1. More detailed information (like temperature, light/dark cycle, salinity, etc) about animal rearing should be added.

Response: Thank you very much. Such data were supplied to the methods.

  1. Furthermore, the authors should clarify the number of individuals per tank and that the results are expressed as the mean of the three replicates, each derived from a pool of 15 samples (if I understand correctly).

Response: Thank you. 15 fish were stocked in each tank. This was stated in the methods.

  1. Likewise, the formulas used for growth performance should be expressed as the mean of three replicates, each derived from a pool of 15 samples. Moreover, reference(s) should be added for the formulas.

Response: thank you. Replications has been expressed in the results section for all parameters measured. Relevant reference was added.

  1. The sentence “two fish were caught for each aquaria and immediately anesthetized…” at line 124 should be changed: it is as if only two fish per group were sampled.

Response: Thank you. Please be noted that we sampled three fish per tank. It was revised.

  1. Finally, references used for the protocols used in “2.4. Digestive enzymes’ assays” are too old and should be updated;

Response: Thank you. The references were updated.

  1. Most of the accession numbers shown in Table 1 has been removed on the gene bank of NCBI

Response: Sorry for the mistake. It seems that during the designing of primers and preparation of the manuscript, the accession numbers were removed. However, this does not affect the present results, as we BLASTED all primers. New accession numbers were added.

  1. The results of gene expression do not seem to be expressed as 2-∆∆CT as stated in Materials and Methods

Response: The results were expressed as 2-∆∆CT and relative to the control group (fold change).

  1. Finally, the Discussion is the real weak point of the paper and it should be rewritten properly. The authors never discuss the growth performance results and most of digestive enzymes and antioxidant activities results.

Response: Thank you. We supplemented this section with new data and tried to connect different results for justification of the observed effects.

  1. Above all, the authors never discuss the results on the basis of the different concentrations administered. It is clear that some data show an increase concentration dependent while others show a decrease from a concentration onward and this should be must be taken into account without generalizing.

Response: Thank you for this comment. We agree you, however, based on the present data we have no ideas why the responses were not similar. To understand this, mechanisms involved in the responses of each parameters must be followed to understand threshold of LA for stimulation each one. We aimed to conclude which LA dose is the suitable one, based on multiple responses. However we mentioned your concern in the conclusion section.

Reviewer 2 Report

animals-2399927

Some positive effects of lactic acid supplementation on the growth and physiological parameters in common carp are demonstrated in this work. Most of analysis were properly executed, and the data are convincing. The manuscript is easy to follow, and the data are clearly presented with some exceptions. However, the authors need to consider following comments to improve their contributions.

L57-59

This might be true in mammals, but not correct for fish since any fish species do not have secretin. Pancreatic digestive enzymes secretion is mainly controlled by cholecystokinin as an endocrine factor. Reconsider this sentence.

L83: “2. Materials and Methods”

An ethical statement is needed.

Table 1

Numbers for annotations after the ingredients (eg. Poultry byproduct”1”) are incorrect.

L122: “Sampling and preservation”

Please mention details of sample conditions.

-       Fasting duration of used samples

-       Were there any digesta in intestine?

Also, anterior intestine was used for pancreatic digestive enzyme assay in this study. However, most of pancreatic tissues are included in hepatopancreas in common carp. Thus, hepatopancreas should be used for digestive enzyme assay rather than intestinal tissue. 

L145: “2.3.2. Digestive enzymes assasy”

Describe the method of analysis for ALP.

L164-167: “Then, a commercial … Bioscience (USA).”

Reconsider this sentence.

Figure 3 and 4: “n = 3”

6 fish were sampled in each diet group. Why only 3 fish used for these assays?

L267-272: “Studies have … organic acids [9, 33].”

Pancreatic digestive enzymes and alkaline phosphatase exhibit their activity in basic condition, so decreased intestinal pH probably reduce the enzyme activities. Please mention author’s opinion for this point.

Author Response

Some positive effects of lactic acid supplementation on the growth and physiological parameters in common carp are demonstrated in this work. Most of analysis were properly executed, and the data are convincing. The manuscript is easy to follow, and the data are clearly presented with some exceptions. However, the authors need to consider following comments to improve their contributions.

L57-59

This might be true in mammals, but not correct for fish since any fish species do not have secretin. Pancreatic digestive enzymes secretion is mainly controlled by cholecystokinin as an endocrine factor. Reconsider this sentence.

Response: Thank you very much. We revised this sentence as “Studies have suggested that organic acids stimulate pancreatic enzymes’ activities in the fish intestine, however, it is not clear if such increases are due to higher secretion of the pancreatic enzymes to lumen and/or derived from microbial secretions in the lumen”

L83: “2. Materials and Methods”

An ethical statement is needed.

Response: Please note that the ethical statement is available at the end of the manuscript.

Table 1

Numbers for annotations after the ingredients (eg. Poultry byproduct”1”) are incorrect.

Response: Thank you. We updated them.

L122: “Sampling and preservation”

Please mention details of sample conditions.  Fasting duration of used samples?  Were there any digesta in intestine?

Response: The fish were fasted 24 h before sampling. The intestine was sampled and analyzed including digesta.

Also, anterior intestine was used for pancreatic digestive enzyme assay in this study. However, most of pancreatic tissues are included in hepatopancreas in common carp. Thus, hepatopancreas should be used for digestive enzyme assay rather than intestinal tissue. 

Response: Thank you. Please note that we did not aim to assess pancreatic hormone production in the pancreas, but the enzymes’ activities in lumen, which can be assumed to improve digestion of nutrients. So we sampled anterior intestine that is the main part of digestion in rainbow trout.

L145: “2.3.2. Digestive enzymes assasy”

Describe the method of analysis for ALP.

Response: Thank you very much. It was added.

L164-167: “Then, a commercial … Bioscience (USA).”

Reconsider this sentence.

Response: Thank you. The sentence was revised.

Figure 3 and 4: “n = 3”

6 fish were sampled in each diet group. Why only 3 fish used for these assays?

Response: Thank you for this comment. First be noted that we sampled three fish per tank (not two), and it is correct in the revised version. “n” is 3, because rearing chambers are considered as replication in aquaculture research. Such sub-samples must not be used for statistical analysis, as it is pseudoreplication and strongly forbidden. So, we used the average of three sub-samples for analysis.

L267-272: “Studies have … organic acids [9, 33].”

Pancreatic digestive enzymes and alkaline phosphatase exhibit their activity in basic condition, so decreased intestinal pH probably reduce the enzyme activities. Please mention author’s opinion for this point.

Response: This sentence has been modified based on your previous comment regarding secretin. However, organic acids do decrease pH in lumen, but not to an acidic pH (below 7). They reduce it slightly, but this may improve breaking of complexes and increase hydrolysis of nutrients.

Reviewer 3 Report

Comments on the manuscript with ID (animals-2399927).

The manuscript contains fundamental errors that cannot be rectified through author revisions. My comments are concentrated only on Table 2 and gene expression results. Suppose the authors double-checked the accession numbers for the genes used in the qRT-PCR study. In that case, they will find that nearly ALL accession numbers were removed from the NCBI GenBank as a result of standard genome annotation processing. Please check these numbers (XM_019074376.1; XM_019126656.1; XM_019100993.1; XM_019111089.1; XM_019106214.1). This information will lead to invalid qPCR results, which is a part of the results presented in this paper. Moreover, to accurately use these primers in this study, the authors must add the results of the gene efficiency studies. This might confirm that these primers worked well in thermocycling and qRT-PCR.

In addition, the results of the gene expression showed that there was OVEREXPRESSION of muc2, muc5, and tnf-α genes over the normal levels, as their values reached 8-, 11-, and 9-fold the control in the 5LA group. These values are too high. Besides, three samples (n=3) in the qPCR results are not enough to accurately demonstrate significant differences between the gene expression results.

Author Response

The manuscript contains fundamental errors that cannot be rectified through author revisions. My comments are concentrated only on Table 2 and gene expression results. Suppose the authors double-checked the accession numbers for the genes used in the qRT-PCR study. In that case, they will find that nearly ALL accession numbers were removed from the NCBI GenBank as a result of standard genome annotation processing. Please check these numbers (XM_019074376.1; XM_019126656.1; XM_019100993.1; XM_019111089.1; XM_019106214.1). This information will lead to invalid qPCR results, which is a part of the results presented in this paper. Moreover, to accurately use these primers in this study, the authors must add the results of the gene efficiency studies. This might confirm that these primers worked well in thermocycling and qRT-PCR.

Response: We understand the concern of the respected reviewer. Please be noted that these accession numbers were valid, when we designed the primers, but may be removed from that time till now. However, this never affects the quality of gene expression data, because we BLASTED the primers to be specific. This means that just wrong accession numbers should be replaced by correct ones, otherwise, the primer sequence is completely correct. We BLASTED them again and confirm all of them. New accession numbers were added to the text. Primer efficiency was also added.

In addition, the results of the gene expression showed that there was OVEREXPRESSION of muc2, muc5, and tnf-α genes over the normal levels, as their values reached 8-, 11-, and 9-fold the control in the 5LA group. These values are too high. Besides, three samples (n=3) in the qPCR results are not enough to accurately demonstrate significant differences between the gene expression results.

Response: Thank you for this comment. I accept that 2-5 folds increase in mRNA levels is common in many studies on fish. However, this does not mean higher expression is not correct, as there is no accepted limits for this. There are several articles from various labs that reported higher expressions over 25 folds, even. In the case of sample size, it should be noted that in aquaculture researches, tanks are units, not fish. So, if we have three tanks per treatment, we will have n = 3. Even, when we samples more fish from a tank, we must use their average for statistical analysis, otherwise, it is pseudo-replication that greatly biases the results.

Round 2

Reviewer 2 Report

Also, anterior intestine was used for pancreatic digestive enzyme assay in this study. However, most of pancreatic tissues are included in hepatopancreas in common carp. Thus, hepatopancreas should be used for digestive enzyme assay rather than intestinal tissue. 

Response: Thank you. Please note that we did not aim to assess pancreatic hormone production in the pancreas, but the enzymes’ activities in lumen, which can be assumed to improve digestion of nutrients. So we sampled anterior intestine that is the main part of digestion in fish. 

→ I did not point out about pancreatic hormone. Pancreatic "digestive enzymes" are produced in the pancreatic tissue. If you want to estimate storage levels of digestive enzymes, hepatopancreas should be used. If you want to secreted enzymes levels, intestinal digesta should be separately analysed.

Figure 3 and 4: “n = 3”

6 fish were sampled in each diet group. Why only 3 fish used for these assays?

Response: Thank you for this comment. First be noted that we sampled three fish per tank (not two), and it is correct in the revised version. “n” is 3, because rearing chambers are considered as replication in aquaculture research. Such sub-samples must not be used for statistical analysis, as it is pseudoreplication and strongly forbidden. So, we used the average of three sub-samples for analysis. 

→ I partially understand this response. In this case, data sets for Figure 1 and 2 should be re-analyzed as "n = 3", and reconsider results and discussion sections too.

Author Response

I did not point out about pancreatic hormone. Pancreatic "digestive enzymes" are produced in the pancreatic tissue. If you want to estimate storage levels of digestive enzymes, hepatopancreas should be used. If you want to secreted enzymes levels, intestinal digesta should be separately analysed.

Response: Thank you for sharing your insight. This makes sense and I will certainly check this in my next study. However, because we aimed to measure a brush boarder enzyme (ALP), we had to use the intestinal tissue, too. So we sampled intestine containing digesta. This is based on several references on common carp; just for example:

https://doi.org/10.1016/j.anifeedsci.2005.09.003

https://doi.org/10.1016/j.aquaculture.2021.737636

https://doi.org/10.1016/j.aquaculture.2019.734656

https://doi.org/10.1016/j.ygcen.2020.113541

I partially understand this response. In this case, data sets for Figure 1 and 2 should be re-analyzed as "n = 3", and reconsider results and discussion sections too.

Response: Please note that we analyzed the data based on "n = 3", but mistakenly stated "n = 6". So, the data and results are relevant. 

Thank you very much for reviewing our manuscript and for your constructive comments.

Reviewer 3 Report

Comments on the manuscript with ID (animals-2399927): -

The authors have responded to the comments raised by the anonymous reviewer. However, there are still minor revisions present in the manuscript.

1. The manuscript needs Extensive English Editing. I suggest sending the manuscript to a native speaker for English Editing and proof-reading.

2. Delete reference [11] as it was published on broiler chickens.

3. Line 630: Oncorhynchus mykiss – write italic.

The manuscript needs Extensive English Editing. I suggest sending the manuscript to a native speaker for English Editing and proof-reading.

Author Response

The manuscript needs Extensive English Editing. I suggest sending the manuscript to a native speaker for English Editing and proof-reading.

Response: Thank you. The manuscript was read and revised by a language teacher and double checked by the Grammarly service.

Delete reference [11] as it was published on broiler chickens.

Response: Thank you. we replaced it with a relevant citation on fish.

Line 630: Oncorhynchus mykiss – write italic.

Response: Thank you. it was revised.